# Design and Synthesis of Aminopyrimidinyl Pyrazole Analogs as PLK1 Inhibitors Using Hybrid 3D-QSAR and Molecular Docking

**DOI:** 10.3390/ph15101170

**Published:** 2022-09-21

**Authors:** Swapnil P. Bhujbal, Hyejin Kim, Hyunah Bae, Jung-Mi Hah

**Affiliations:** 1Department of Pharmacy, College of Pharmacy, Hanyang University, Ansan 426-791, Korea; 2Institute of Pharmaceutical Science and Technology, Hanyang University, Ansan 426-791, Korea

**Keywords:** cancer, PLK1, kinase, 3D-QSAR, molecular docking, inhibitors

## Abstract

Cancer continues to be one of the world’s most severe public health issues. Polo-like kinase 1 (PLK1) is one of the most studied members of the polo-like kinase subfamily of serine/threonine protein kinases. PLK1 is a key mitotic regulator responsible for cell cycle processes, such as mitosis initiation, bipolar mitotic spindle formation, centrosome maturation, the metaphase to anaphase transition, and mitotic exit, whose overexpression is often associated with oncogenesis. Moreover, it is also involved in DNA damage response, autophagy, cytokine signaling, and apoptosis. Due to its fundamental role in cell cycle regulation, PLK1 has been linked to various types of cancer onset and progression, such as lung, colon, prostate, ovary, breast cancer, melanoma, and AML. Hence, PLK1 is recognized as a critical therapeutic target in the treatment of various proliferative diseases. PLK1 inhibitors developed in recent years have been researched and studied through clinical trials; however, most of them have failed because of their toxicity and poor therapeutic response. To design more potent and selective PLK1 inhibitors, we performed a receptor-based hybrid 3D-QSAR study of two datasets, possessing similar common scaffolds. The developed hybrid CoMFA (*q^2^* = 0.628, *r^2^* = 0.905) and CoMSIA (*q^2^* = 0.580, *r^2^* = 0.895) models showed admissible statistical results. Comprehensive, molecular docking of one of the most active compounds from the dataset and hybrid 3D-QSAR studies revealed important active site residues of PLK1 and requisite structural characteristics of ligand to design potent PLK1 inhibitors. Based on this information, we have proposed approximately 38 PLK1 inhibitors. The newly designed PLK1 inhibitors showed higher activity (predicted pIC_50_) than the most active compounds of all the derivatives selected for this study. We selected and synthesized two compounds, which were ultimately found to possess good IC_50_ values. Our design strategy provides insight into development of potent and selective PLK1 inhibitors.

## 1. Introduction

The genetic stability of all eukaryotes is governed by the flawless segregation of chromosomes during mitosis. Disruption of this phenomenon can lead to aneuploidy, which is a vital cause of cancer [1]. Cancer continues to be one of the most serious public health problems worldwide. As per the statistics of WHO, cancer is the second major cause of death worldwide, with approximately 9.6 million deaths in 2018. Men are more likely to develop prostate, lung, colorectal, stomach, and liver cancer than women, who are more likely to develop colorectal, breast, lung, thyroid, and cervical cancer. Protein kinase inhibitors developed to treat various cancer types have been of great advantage in recent years [1]. However, these inhibitors are merely efficient and are limited by the emergence of resistant mutations [1,2]. Interestingly, polo-like kinase 1 (PLK1), which is an important type of serine-threonine kinase, controls several phases of mitosis [1,3]. PLK1 is a primary regulator of mitotic progression, the overexpression of which is often related to oncogenesis and is an important therapeutic target for anticancer drug discovery [4]. 

There are five isoforms of PLKs in humans, referred to as PLK1, PLK2, PLK3, PLK4, and PLK5 [5,6]. PLK1 is highly characterized among them and has the utmost degree of homology with the single PLK gene [5]. PLK1 is structurally defined by the presence of 3 functional domains; a disordered and highly conserved N-terminal domain, C-terminal polo-box domain (PBD) for substrate targeting and implied in its subcellular localization, and a kinase domain (KD) that is regulated through phosphorylation by upstream kinases [7,8,9]. The N-terminal domain is a Ser/Thr kinase domain that has a T-loop, whose phosphorylation is directly associated with the PLK1’s kinase activity [6]. The PBD is C-terminal domain, made of two polo-box structures, which is activated upon ligand binding and separates with T-loop of the kinase domain [6,10].

PLK1 is predominantly related to regulation of the cell cycle, disruption of which is the primary cause of cancer [10]. In addition to its role in the activity of tumor suppressors and oncogenes, PLK1 has a distinctive function in regulating cancer cell metabolism, promoting the growth of cancer cells [5]. It also has an indisputable role in controlling several key transcription factors that promote cell proliferation, transformation, and epithelial-to-mesenchymal transition. PLK1 expression begins to increase from the S/G2 phase and peaks at mitosis [9]. PLK1 activates the cyclin B/cdc2 complex by phosphorylating CDC25, which stimulates cell proliferation [5,6]. A few recent studies have reported that PLK1 also plays role in autophagy, chromosomal stability, and DNA damage response. Considering the crucial role of PLK1 during DNA damage repair and cell-cycle regulation, it is not astonishing that it is upregulated in various types of cancers, such as melanoma, colorectal cancer, non-small cell lung cancer, thyroid carcinoma, esophageal carcinoma, ovarian carcinoma, colorectal cancer, breast cancer, and prostate cancer [3,4,5]. Moreover, PLK1 is linked to diverse immune and neurological disorders, such as graft versus host disease, liver fibrosis, Huntington’s disease, and Alzheimer’s disease [11,12,13].

Consequently, inhibition of PLK1 has been an appealing goal for researchers, who have made substantial efforts to design and develop small molecule PLK1 inhibitors (Figure 1). The FDA has approved at least 55 small molecule anticancer kinase inhibitors as of February 2020 [14]. Researchers from Boehringer Ingelheim developed one of the earliest PLK1 inhibitors, BI2536, a dihydropteridinone derivative. It was the first pioneered PLK1 inhibitor to enter clinical trials for the treatment of cancer, however it was not very effective during monotherapy regimens in clinical trials [5,10,14]. One more Boehringer Ingelheim compound, volasertib (aka BI6727), is an ATP-competitive inhibitor that was developed by modifying the chemical structure of BI2536 and was shown to inhibit the growth of acute myeloid leukemia (AML) cell lines. However, when combined with Trematinib, BI6727 was more effective at arresting dual G1 and G2-M in an NRAS-mutant melanoma cell line [15]. Both of these drugs passed phase I and III clinical trials for breast cancer, B-cell lymphoma and AML but were found to be more effective in combination with other drugs and are no longer used in monotherapy [5,16].

Onvansertib is a 3rd generation, highly selective and oral PLK1 NCD (N-terminal catalytic domain) inhibitor that exhibited promising synergic results during clinical trials as part of a combination regimen treatment of AML and metastatic colorectal cancer [6,15]. Currently, it is the only PLK1 inhibitor under clinical trials for solid tumors. In addition, Rigosertib is presently in Phase III clinical trials for treatment of cancers, including AML. Nevertheless, it is a non ATP-competitive dual inhibitor of PI3K and PLK1 [6,15]. Furthermore, Phase I clinical trials for two PLK1 NCD inhibitors, GSK461364, and Tak960, were finished in 2009 and 2013, respectively, but did not undergo further development [16]. PLK1 inhibitor, NMS-P937 (pyrazoloquinazoline derivative), has shown remarkable reduction of osteosarcoma tumor growth. NMS-P937 is under phase 1 trials for patients having advanced metastatic solid tumors, nevertheless, no findings have yet been published [16,17]. 

In summary, there are numerous small molecule PLK1 inhibitors in research and in the early stages of clinical progress. Nonetheless, most of these convincing drugs targeting PLK1 have been inefficient in clinical trials due to their toxicity leading to serious side effects or low therapeutic response [15,17]. PLK1 inhibitors that are highly selective are anticipated to overcome side effects impelled by off-target effects. An effective way is to design selective PLK1 inhibitors by using structure-based drug design methods, as in our study. We selected two datasets as PLK1 inhibitors with similar scaffolds, such as pyrimidine derivatives [3] and quinazoline derivatives [4]. We chose compound **17** (Appendix A) from dataset 1 as a reference compound (as shown in Figure 2) to combine the remaining compounds from both datasets for hybrid 3D-QSAR because a detailed mechanistic study of compound **17** showed that PLK1 inhibition by **17** enhanced mitotic arrest at the G2/M phase checkpoint and led to apoptosis of cancer cells. Compound **17** is also one of the most active compounds among datasets 1 and 2.

The docking of compound **17** into the active site of PLK1 revealed important interactions with crucial binding site residues, which were accountable for inhibition of PLK1. Combining the specific structural information from 3D-QSAR contour maps with the overall molecular docking analysis about ligand–protein interactions provided insight toward understanding and modifying pyrimidine scaffold of PLK1 inhibitors for better potency. Hence, we designed few PLK1 inhibitors of aminopyrimidinyl scaffold, which showed better predicted activity (pIC_50_) than the most active compound in the datasets used in our study. Among our designed compounds, we synthesized two compounds for experimental validation, which showed good inhibitory activity (IC_50_). Our designed strategy, which was validated by experimental study could be a great start to develop potent and selective PLK1 inhibitors for medicinal chemists and pharmaceutical companies.

## 2. Results and Discussion

### 2.1. Molecular Docking

Molecular docking was performed to interpret the binding mode of the most active compound, **17**, inside the active site of PLK1. We utilized an extended sampling protocol to perform induced fit docking that produced 80 binding poses for compound **17**. All 80 poses were checked for a docking score and bonding and non-bonding interactions. One of the poses displayed a docking score of −12.04 and yielded a binding pose with PLK1 that was similar to that witnessed between the co-crystallized ligand (BI6727) and the protein (PDB ID: 3FC2) (Figure 3). This pose was chosen to analyze further interactions. The most active compound **17** docked within the binding pocket, mimicked the binding mode of ATP, and formed three hydrogen bonds (H-bonds) with crucial active site residues of PLK1. The hydrogen and nitrogen atoms from the aminopyrimidine ring of compound **17** formed 2 H-bonds with the key hinge region residue CYS133. This interaction was also observed in previously reported docking studies of several other PLK1 inhibitors including BI6727, which was considered important for inhibition of PLK1 [3,4]. Another H-bond was found between the oxygen atom from the dimethylamino-propan-2-one moiety at R^3^ position and the residue SER137. This moiety docked very well within a pocket lined primarily with residues of the kinase C-lobe, including SER137 that formed the floor of the C-lobe [4]. In addition, pyrimidine ring formed pi–pi interaction with GLU131, which is also a part of the hinge region. Overall interactions were alike to those observed between a co-crystallized ligand and PLK1. Hence, the docked pose of compound **17** validated a strong binding conformation.

The docked pose of compound **17** was further analyzed to assess hydrophobic interactions. A Python script ‘color h’ was utilized to color the hydrophobic residues of PLK1 and to identify their interactions with compound **17**. PyMOL uses this script and Eisenberg hydrophobicity scale (Figure 3b) to color the receptor [18]. The most hydrophobic residues were colored red, whereas the least hydrophobic residues were colored white. The substituents, trifluoromethylpyrimidine at the R^2^ position and methylthiophene-2-carboxylate at the R^1^ position of the ligand, were docked within the hydrophobic pocket lined by hydrophobic residues CYS67, ALA80, LYS82, LEU59, and LEU130. Additionally, the dimethylamine group at R^3^ position formed hydrophobic interactions with LEU139 and GLY180. The hydrophobic residues CYS67, ALA80, LYS82, and LEU130 were crucial in the hydrophobic pocket, since their interaction with compound BI6727 (co-crystallized ligand) was observed. Comprehensive docking analysis suggested that the selected docked pose of the most potent compound **17** was appropriate, thus it was employed to further perform 3D-QSAR studies.

### 2.2. Hybrid 3D-QSAR Models

We obtained the receptor-based hybrid 3D-QSAR models (CoMFA and CoMSIA) after combining two datasets with pyrimidine and quinazoline scaffolds. These two datasets were combined in order to acquire and understand the structural characteristics requisite to propose more potent PLK1 inhibitors and to study structure–activity relationships. The docked pose of the most active compound, compound **17** was selected as a reference compound to align the remaining compounds of the dataset using a common scaffold alignment method (Figure 4), which provided better statistical CoMFA and CoMSIA models in SYBYL-X 2.1. The dataset was separated into a training set of 52 compounds and a test set of 18 compounds using the criteria proposed in activity ranking algorithm 4 in a previously reported article [19]. Accordingly, our test set contained compounds with low, moderate, and high activity (pIC_50_) values. The test set compounds are specified by * in Appendix A entitled the chemical structures of the selected PLK1 inhibitors with their IC_50_ values.

The reliability of the developed hybrid 3D-QSAR models was examined by computing different statistical parameters, such as the non-cross validated correlation coefficient (*r^2^*), cross-validated correlation coefficient (*q^2^*), standard error of estimate (SEE), F-value, and the optimal number of components (ONC) with the help of partial least square (PLS) analysis. Initially, hybrid CoMFA (*q^2^* = 0.517, ONC = 6, *r^2^* = 0.847) and CoMSIA (*q^2^* = 0.540, ONC = 6, *r^2^* = 0.855) models were obtained for the complete dataset compounds (training + test set), which were called full models. Test set 12 was used to develop CoMFA (*q^2^* = 0.628, ONC = 6, *r^2^* = 0.905) and CoMSIA (*q^2^* = 0.580, ONC = 6, *r^2^* = 0.895) models. CoMFA models were generated using steric and electrostatic fields, while CoMSIA employed hydrogen bond acceptor, donor, and hydrophobic fields along with steric and electrostatic fields. Thus, CoMSIA models were produced using various combinations of these fields (Appendix A). The model with the best *q^2^* and *r^2^* values was chosen as the concluding model. The combination of steric, electrostatic, and hydrophobic fields yielded reasonable CoMSIA model. In conclusion, CoMFA and CoMSIA models obtained using external test set 12 were selected for further statistical analysis. Several methods were used to validate these models. Table 1 provides thorough statistical values of the validated CoMFA and CoMSIA models.

#### Validation of 3D-QSAR Models

The selected hybrid 3D-QSAR models were validated using a number of validation techniques to evaluate predictive ability and robustness. Validation techniques of predictive *r^2^ (r^2^pred)*, bootstrapping, leave-out-five (LOF), and *rm^2^* metric calculations presented acceptable statistical values [20,21]. Hence, the generated models were robust and predictive. Table 1 depicts the detailed statistical values. Residual values as well as the experimental and predicted activity values of the selected CoMFA and CoMSIA models can be seen in Appendix A. The scatter plots for the same are depicted in Figure 5.

### 2.3. Contour Map Analysis

#### 2.3.1. CoMFA Contour Maps

The contour maps of the hybrid CoMFA model are depicted superimposed with those of compound **17** in Figure 6. In the steric contour map (Figure 6A), green and yellow colored contour maps revealed favorable and unfavorable regions for steric substitution, respectively. In Figure 6B (electrostatic contour map), the blue contour denoted a favorable region for electropositive substitution, whereas the red contour denoted unfavorable regions. 

Two green colored contours were observed at the R^1^ and R^3^ positions, illustrating that bulky groups were favorable at these positions to increase potency. The R^1^ position that holds a steric group could interact with the hydrophobic residues of PLK1. This could be signified by hydrophobic interactions of methylthiophene-2-carboxylate with residues GLY60 and LEU130, which were also identified in the docking analysis of compound **17**. Conversely, two yellow contour maps were spotted near the indoline ring at the R^3^ position, which conveyed the inferiority of bulky groups at this position. There was also a small yellow contour near the methyl acetate moiety at the R^1^ position, indicating that addition of bulky groups at this position might not increase potency.

Furthermore, red and blue contours were observed near the aminopyrimidine ring at the R^3^ position among which, red contours seemed to be position specific. Therefore, the presence of blue contours at this position showed that electropositive groups were favorable, which could be verified by looking at key hydrogen bond interactions of amine groups with the hinge region residue CYS133 in the docking study of compound **17**. Another red contour was seen near the indoline ring at the R^3^ position, which meant that electronegative substitution was favorable at this position. This was verified with a docking study of compound **17** where H-bond interactions were found between the oxygen atom from dimethylamino-propan-2-one moiety and the residue SER137. Compounds **13, 33**, and the most active compound **17**, which have moderate to high activity, could be the result of this. 

#### 2.3.2. CoMSIA Contour Maps

The combination of steric, electrostatic and hydrophobic (SEH) fields was utilized to produce CoMSIA contour maps (Figure 7). The steric and electrostatic contour maps are quite similar to the CoMFA steric and electrostatic contours, thus only a hydrophobic contour map is discussed below. The hydrophobic contour map is displayed in the Figure 7C, in which magenta contours represent favorable areas for hydrophobic substitution, however cyan contours denote unfavorable regions. 

Two big magenta contours are present at R^1^ and R^2^ positions, which illustrates that occurrence of hydrophobic group at this position could increase the activity of the compound. This can be proved by hydrophobic interactions of trifluoromethylpyrimidine and methylthiophene-2-carboxylate with residues GLY60, CYS67, ALA80, LYS82, LEU59, and LEU130 that were seen in our docking analysis of the compound **17**. Moreover, two cyan contour maps are present near the R^3^ position, which explain that substituting hydrophobic groups at this location can reduce the activity of a compound.

### 2.4. Designing New PLK1 Inhibitors

The created 3D-QSAR models revealed crucial structural properties in terms of steric, electrostatic, and hydrophobic fields. These structural characteristics and the important interactions detected in the docking analysis of the most potent compound **17** were utilized to derive a drug design strategy with a new scaffold to design new PLK1 inhibitors (Figure 8). As per this design strategy, we modified the substituent at the R^1^ position, while keeping others fixed to assess differences in activity. The same procedure was followed for each position and identified new PLK1 inhibitors. The methoxy group at the R^1^ position, dimethylacetamide at the R^2^ position, CF_3_ at the R^3^ position, amide (CONH_2_) at the R^4^ position, and chlorine at the R^6^ position showed better activity (pIC_50_) than the most potent compound **17** in the dataset.

The structures and the predicted pIC_50_ values of the newly designed compounds are presented in Appendix A. A few selected PLK1 inhibitors are shown below in Table 2.

#### Synthesis of New PLK1 Inhibitors and Evaluation of IC_50_ Values

Depending on the their fastest synthetic route in order to validate our PLK1 drug design, we selected two compounds for the synthesis: compound D39 [ethyl €-1-(2-((1-(dimethylcarbamoyl)indolin-6-yl)amino)pyrimidin-4-yl)-4-styryl-1H-pyrazole-3-carboxylate] and compound D40 [(E)-6-((4-(3-carbamoyl-4-styryl-1H-pyrazol-1-yl)pyrimidin-2-yl)amino)-*N,N*-dimethylindoline-1-carboxamide]. NMR and HRMS data for these compounds are added in the methodology section. Synthesis of the rest of the compounds is under process, which upon completion will be published as a separate article with detailed procedure and experimental evaluations. Furthermore, we docked these two compounds inside the active site of PLK1 to check their binding mode (Appendix A). Both compounds formed two key hydrogen bonds with hinge region residue CYS133 similar to compound **17**. Compound D39 also formed pi–pi and pi–cation interactions with residues PHE183 and LYS82, respectively. Furthermore, a hydrogen from amide group at R^4^ position of compound D40 formed additional hydrogen bond with residue ASP194 that is considered to be crucial for the selectivity over other kinases. It also possessed pi–pi interaction with PHE183. 

Moreover, IC_50_ evaluation of compounds D39 and D40 revealed that they possess good inhibitory activities of 1.43 µM and 0.359 µM, respectively (Table 3). These results validates our designed compounds and design scheme that could be further utilized to develop more potent PLK1 inhibitors.

## 3. Materials and Methods

### 3.1. Training Set/Test Set Selection for CoMFA and CoMSIA

We selected two datasets comprising pyrimidine derivatives [3] and quinazoline derivatives [4] as PLK1 inhibitors for the hybrid 3D-QSAR study. Dataset 1 and dataset 2 consisted of 35 and 39 compounds, respectively, exhibiting a log value of greater than 3.5 logarithmic units, which was within the required range [22]. Compound structures were drawn using the sketch module in SYBYL-X 2.1 and were optimized using energy minimization with Tripos force field [23]. Biological activities (IC_50_) of all compounds in the study were converted into pIC_50_ (−log IC_50_) values, which were used as dependent variables to develop 3D-QSAR models. The compounds from both the datasets were divided into training set of 52 compounds for model generation and 18 compounds as test set for model validation. The structure and activity of the compounds were considered in order to separate them into training and test sets. The compounds with low, medium, and high activity values were carefully added to the test set as suggested in Algorithm 4 [19]. Compounds with undefined activity values were eliminated as outliers. The chemical structures of the selected dataset compounds with their IC_50_ values are depicted in Appendix A. 

### 3.2. Molecular Docking 

Molecular docking of the most active compound **17** from the selected dataset was performed using Schrodinger Maestro 12.8 (Release 2021-2, Schrödinger, LLC, New York, NY, USA) [24]. The structure of compound **17** was drawn using Chemdraw [25] and its 3D conformation was generated using the Schrödinger LigPrep program [26]. LigPrep produced all probable tautomers and states at pH 7.0 using Epik [27] for compound **17,** and specified chiralities were retained following minimization using the OPLS 2005 force field [28]. The crystal structure of PLK1 co-crystallized with BI6727 (PDB ID: 3FC2) was taken from the Protein Data Bank (PDB). The Protein Preparation Wizard was utilized to prepare protein by assigning bond orders, hydrogens at pH 7.0, and removing water molecules [29]. Prime was used to complete missing side chains and loops. Finally, a restrained minimization was performed using the default constraint of 0.30 Å RMSD and the OPLS 2005 force field to finalize the protein preparation. Molecular docking simulations were executed with the help of a Glide induced fit docking module in extended sampling protocol mode [30]. The docked conformations of compound **17** were examined to identify important interactions with the active site residues of PLK1. The selected docked pose of compound **17** was employed as a template to align the rest of the dataset compounds for 3D-QSAR model generation. 

### 3.3. Receptor-Based Hybrid CoMFA and CoMSIA Models

SYBYL-X 2.1 [31] was used to develop 3D-QSAR (3-Dimensional Quantitative Structure–Activity Relationship) models utilizing CoMFA (Comparative Molecular Field Analysis) [32] and CoMSIA (Comparative Molecular Similarity Indices Analysis) [33] to correlate 3D structures of the PLK1 inhibitors with the biological activity. The alignment of dataset compounds was performed inside the active site of the receptor using a common scaffold alignment method using the most active compound **17** as a template molecule. In CoMFA, the steric and electrostatic potential energies were estimated using Lennard–Jones and Coulombic potentials, respectively [31].

The application of appropriate partial charge was crucial toward obtaining reasonable 3D-QSAR models. We used Gastegeir Marsili as a partial charge scheme and default parameters to generate 3D-QSAR models [34]. A grid spacing of 2.0 Å and an sp^3^ hybridized carbon as a probe atom with +1 charge were used. Statistically acceptable CoMFA and CoMSIA models were obtained using partial least squares (PLS) regression. CoMFA descriptors as independent variables and biological activity values (pIC_50_) as dependent variables were used in PLS regression. The reliability of the generated models was assessed through PLS analysis with leave-one-out (LOO) cross-validation and to calculate the squared cross-validated correlation coefficient (*q^2^*) value, an optimal number of components (ONC) and the standard deviation of prediction (SEP). A column filtering value of 2.0 and obtained ONC were used in non-cross-validation analysis to compute the squared correlation coefficient (*r^2^*), F-test value (F), and standard error of the estimate (SEE). 

Nevertheless, CoMSIA utilizes descriptors, such as steric, electrostatic, hydrophobic, hydrogen bond acceptor, and donor. All of these CoMSIA similarity indices were calculated using a probe atom of radius 1.0 Å and an attenuation factor of 0.30. A Gaussian function was used to calculate the CoMSIA model between the grid point and each atom of the molecule. Various CoMSIA models were derived based on different descriptor combinations using the same lattice box that was used in CoMFA. The model that exhibited acceptable *q^2^* and *r^2^* statistical values was chosen as the final model. Additionally, the selected models were statistically validated using the following validation methods.

### 3D-QSAR Model Validation 

Several validation techniques, including bootstrapping, leave-out-five (LOF), *rm^2^* metric calculation, and external test set validation were implemented to assess the stability, robustness, and predictive ability of the resulting models. To evaluate model’s reliability, bootstrapping for 100 runs and progressive scrambling of 10 samplings with 2–10 bins were performed [35]. Last, the predictive ability was calculated as expressed through the predictive correlation coefficient (*r^2^_pred_*), using the formula given below:*r^2^_pred_* = (SD − PRESS)/SD
where SD is the sum of the squared deviations of each experimental value from the mean and PRESS is the sum of the squared differences between the predicted and actual affinity values. 

Standard contour maps were developed for both CoMFA and CoMSIA models. A new design strategy was derived using the structural information from an analysis of the contour maps and molecular docking, and we designed more potent PLK1 inhibitors.

### 3.4. Synthesized PLK1 Inhibitors

We selected two compounds for synthesis to validate our design of PLK1 inhibitor: compound D39, Ethyl (*E*)-1-(2-((1-(dimethylcarbamoyl)indolin-6-yl)amino)pyrimidin-4-yl)-4-styryl-*1H*-pyrazole-3-carboxylate and compound D40, (*E*)-6-((4-(3-carbamoyl-4-styryl-*1H*-pyrazol-1-yl)pyrimidin-2-yl)amino)-*N,N*-dimethylindoline-1-carboxamide. We state the spectral data of the two synthesized compounds below in Table 4. 

### 3.5. Evaluation of IC_50_ Values

We used Reaction Biology Corp. Kinase Hot Spot^SM^ service (www.reactionbiology.com, accessed date 11 January 2022) for screening of D39 and D40 (10 µM) and IC_50_ Profiler Express for IC_50_ measurement. Assay protocol: In a final reaction volume of 25 µL, substrate [Casein], 1 µM, ATP 10 µM, PLK1 (h) (5–10 mU) is incubated with 25 mM Tris pH 7.5, 0.02 mM EGTA, 0.66 mg/mL myelin basic protein, 10 mM Mg acetate, and [^33^P-ATP] (specific activity approximately 500 cpm/pmol, concentration as required). The reaction is initiated by addition of the Mg-ATP mix. After incubation for 40 min at room temperature, the reaction is stopped by addition of 5 µL of a 3% phosphoric acid solution. Next, 10 µL of the reaction is spotted onto a P30 filtremat and washed three times for 5 min in 75 mM phosphoric acid and once in methanol prior to drying and scintillation counting.

## 4. Conclusions

One of the most important cell cycle regulators, PLK1, a type of serine-threonine kinase has garnered the attention of the academic research community as well as pharmaceutical companies to develop anticancer inhibitors. PLK1 regulates the cell cycle, and cell cycle dysregulation is the primary cause of cancer. In addition to its role in mitosis, PLK1 has a unique function in regulating cancer cell metabolism, which promotes the growth of cancer cells. Therefore, it is imperative to design and develop new PLK1 inhibitors. Several PLK1 inhibitors were discovered in the past few years; some were unsuccessful due to their low therapeutic response and side effects. Some of these inhibitors, such as volasertib, Onvansertib, etc., were in use for treatment of AML, prostate cancer, and thyroid cancer but were later found to be more effective in combination with other drugs and are no longer used in monotherapy. Hence, we focused on designing more potent PLK1 inhibitors through receptor-based hybrid 3D-QSAR and molecular docking studies. Molecular docking of compound **17** revealed important active site residues of PLK1, such as CYS133, SER137, CYS67, ALA80, LYS82, LEU59, and LEU130. The generated hybrid CoMFA (*q^2^* = 0.628, ONC = 6, *r^2^* = 0.905) and CoMSIA (*q^2^* = 0.580, ONC = 6, *r^2^* = 0.895) models possessed acceptable statistical results and provided very useful structural information to modify the pyrimidine and quinazoline scaffolds used in this study. Hence, we developed a design strategy and identified additional potent PLK1 antagonists. Our designed compounds showed better predicted activity (pIC_50_) than the most active compound in the dataset. Altogether, results of our study provided crucial structural insights to develop and synthesize more selective and potent PLK1 inhibitors. Therefore, we synthesized and evaluated IC_50_ values of two designed compounds D39 and D40. The IC_50_ value of compound D39 and D40 was found to be 1.43 µM and 0.359 µM, respectively. Our designed strategy and designed inhibitors can be used as a reference by the drug design community to develop more potent PLK1 antagonists.

## Figures and Tables

**Figure 1 pharmaceuticals-15-01170-f001:**
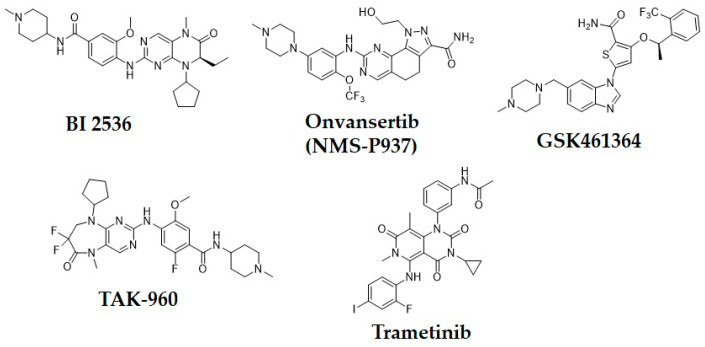
Chemical structures of the previously reported PLK1 inhibitors.

**Figure 2 pharmaceuticals-15-01170-f002:**
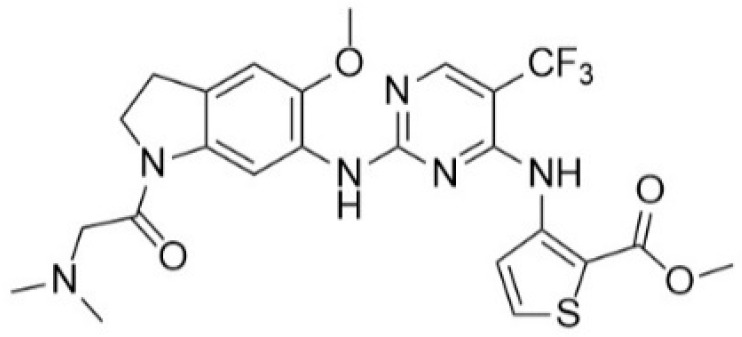
Chemical structure of Compound **17** (The most active compound used in our study).

**Figure 3 pharmaceuticals-15-01170-f003:**
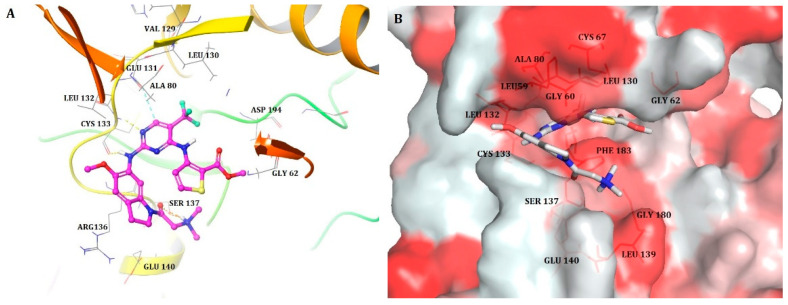
(**A**) Docked pose of the most active compound, **17**, within the active site of PLK1 (Hydrogen bonds are represented as yellow dotted lines, pi-pi interactions are represented as cyan dotted lines); (**B**) The most active compound **17** (shown in stick model) within the hydrophobic pocket of PLK1; the red colored region represents the most hydrophobic surface of the protein, and the white color represents the least hydrophobic surface. Hydrophobic residues are indicated with red lines.

**Figure 4 pharmaceuticals-15-01170-f004:**
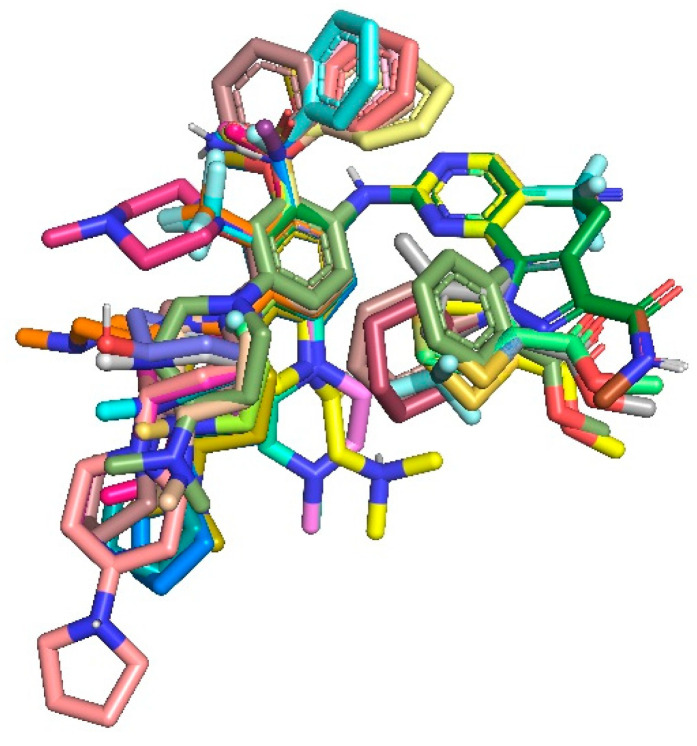
Alignment of the dataset compounds for hybrid 3D-QSAR.

**Figure 5 pharmaceuticals-15-01170-f005:**
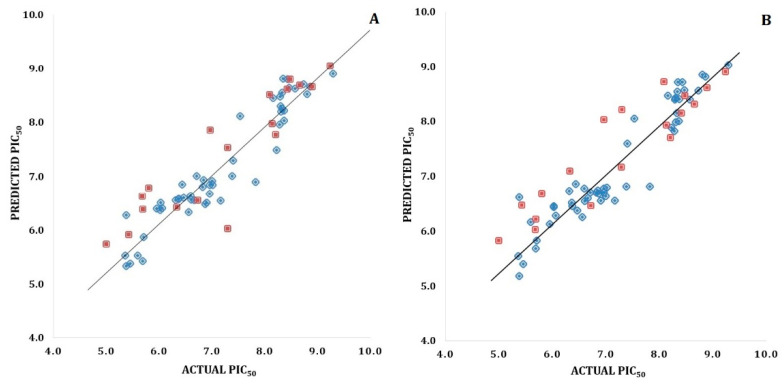
(**A**) Scatter plot for the selected CoMFA model; (**B**) Scatter plot for the selected CoMSIA model; the plot shows the actual pIC_50_ versus predicted pIC_50_ activity of the training and test sets; the training set compounds are represented as blue diamonds; the test set compounds are represented as dark red squares.

**Figure 6 pharmaceuticals-15-01170-f006:**
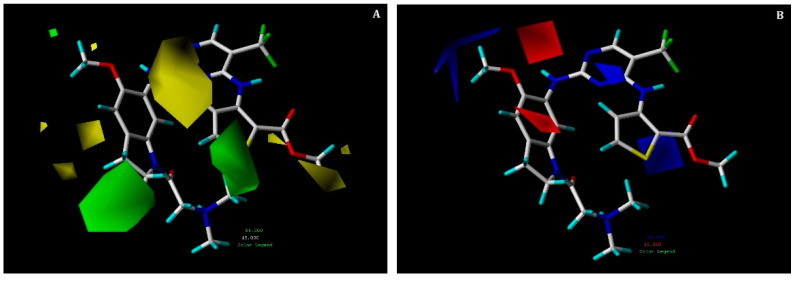
Contour maps for the selected CoMFA model. (**A**) Steric contour map; (**B**) electrostatic contour map; green contours show the region is favorable for bulky substitutions, and yellow contours show the region is unfavorable; blue contours favor electropositive substitutions, whereas red contours favor electronegative substitutions.

**Figure 7 pharmaceuticals-15-01170-f007:**
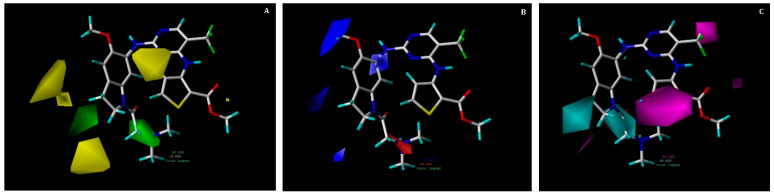
Contour maps for the selected CoMSIA model. (**A**) Steric contour map; (**B**) electrostatic contour map; (**C**) hydrophobic contour map. Green contours show regions that are favorable for bulky substitutions, whereas yellow contours show unfavorable regions; blue contours favor electropositive, while red does not. Magenta contours show the regions favorable for hydrophobic substitutions, whereas cyan contours show unfavorable regions.

**Figure 8 pharmaceuticals-15-01170-f008:**
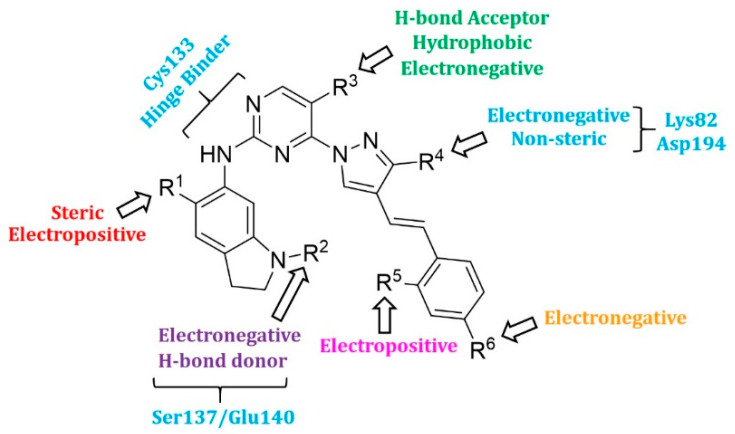
The design strategy for PLK1 inhibitors obtained from hybrid 3D-QSAR models.

**Table 1 pharmaceuticals-15-01170-t001:** Detailed statistical values of the selected CoMFA and CoMSIA models.

Parameter	Full Model		Test Set 12	
	CoMFA	CoMSIA (SEH)	CoMFA	CoMSIA (SEH)
*q^2^*	0.517	0.540	0.628	0.580
ONC	6	6	6	6
SEP	0.844	0.824	0.717	0.762
*r^2^*	0.847	0.855	0.905	0.895
SEE	0.475	0.462	0.363	0.381
F value	58.087	61.993	71.401	63.990
LOF	-	-	0.607	0.609
BS-*r^2^*	-	-	0.929	0.936
BS-SD	-	-	0.020	0.020
*r^2^pred*	-	-	0.796	0.783
*rm^2^*	-	-	0.665	0.581
Delta *rm^2^*	-	-	0.181	0.214

*q^2^*: squared cross-validated correlation coefficient; ONC: optimal number of components; SEP: standard error of prediction; *r^2^*: squared correlation coefficient; SEE: standard error of estimation; F value: F-test value; LOF: leave-out-five; BS-*r^2^*: bootstrapping *r^2^* mean; BS-SD: bootstrapping standard deviation; *r^2^_pred_*: predictive *r^2^; rm^2^:* average *rm^2^* metric calculation; Delta *rm^2^*: standard error.

**Table 2 pharmaceuticals-15-01170-t002:** The structures and predicted pIC_50_ values of few selected designed PLK1 inhibitors.

Compound Structure	Name	R^1^	R^2^	R^3^	R^4^	R^5^	R^6^	Predicted pIC_50_
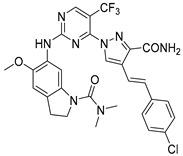	D3	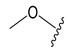	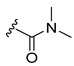	CF_3_	CONH_2_	H	Cl	10.217
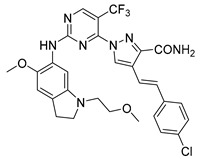	D5	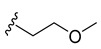	9.715
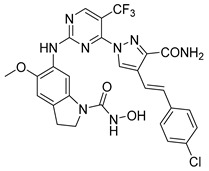	D10	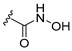	10.272
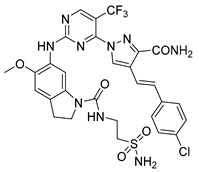	D14	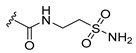	9.811
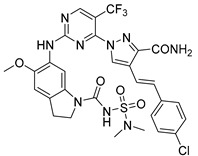	D17	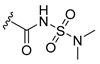	9.903

**Table 3 pharmaceuticals-15-01170-t003:** The comparison of the actual and calculated IC_50_ values of two synthesized PLK1 inhibitors.

Compound Structure	Name	R^1^	R^2^	R^3^	R^4^	R^5^	R^6^	IC_50_ (µM)	Predicted IC_50_ (nM)
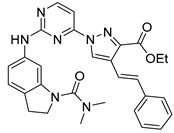	D39	H	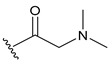	H	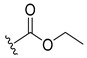	H	H	1.43	0.35
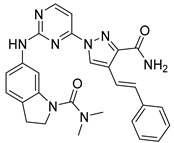	D40	H	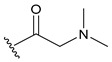	H	CONH_2_	H	H	0.359	0.13

**Table 4 pharmaceuticals-15-01170-t004:** The spectral data (NMR and HRMS) of the synthesized compounds.

Compound	Structure	NMR	HRMS
D39	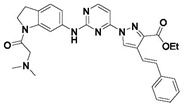	^1^**H NMR** (400 MHz, DMSO-*d_6_*) δ 10.05–9.95 (m, 1H), 9.40 (s, 1H), 0.16 (d, *J* = 28.1 Hz, 1H), 8.62 (d, *J* = 5.3 Hz, 1H), 7.58 (d, *J* = 7.3 Hz, 1H), 7.48 (d, *J* = 1. 8 Hz, 1H), 7.41 (t, *J* = 7.6Hz, 1H), 7.33 (dd, *J* = 5.0, 3.6 Hz, 1H), 7.31–7.09 (m, 5H), 4.40 (q, *J* = 7.1 Hz, 2H), 4.31–4.23 (m, 3H), 4.10 (q, *J* = 5.2 Hz, 2H), 3.22 (s, 1H), 2.27 (d, *J* = 16.0 Hz, 1H), 2.11 (d, *J* = 23. 6 Hz, 6H), 1.38 (t, *J* = 7.1 Hz, 2H); **^13^C NMR** (101 MHz, DMSO-*d_6_*) δ 168.4 (S), 161.7 (s), 161.2 (s), 159.6 (s), 156.5 (s), 143.3 (s), 142.9 (s), 138.9 (s), 137.2 (s), 131.6 (s), 128.8 (s), 127.9 (s), 126.5 (s), 125.9 (s), 125.1 (s), 124.5 (s), 124.3 (s), 117.5 (s), 114.5 (s), 107.1 (s), 99.1 (s), 63.9 (S), 61.0 (s), 47.9 (s), 45.2 (s), 27.1 (s), 14.2 (s)	**HRMS (ESI+)** calculated for [M + H]^+^ C_30_H_31_N_7_O_3_: 538.2561, found 538.2572.
D40	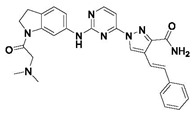	**^1^H NMR** (400 MHz, DMSO-*d_6_*) δ 9.99 (s, 1H), 9.36 (s, 1H), 9.23 (s, 1H), 8.63 (d, *J* = 5.3 Hz, 1H), 7.97 (s, 1H), 7.63 (d, *J* = 16.7 Hz, 1H), 7.59 (s, 1H), 7.55 (d, *J* = 7.4 Hz, 2H), 7.48 (d, *J* = 2.8 Hz, 1H), 7.47 (d, *J* = 5.3 Hz, 1H), 7.40 (t, *J* = 7.6 Hz, 2H), 7.29 (t, *J* = 7.3 Hz, 1H), 7.17 (d, *J* = 8.1 Hz, 1H), 7.07 (d, *J* = 7.8 Hz, 1H), 4.27 (t, *J* = 8.4 Hz, 2H), 3.24 (s, 2H), 3.09 (t, *J* = 8.3 Hz, 2H), 2.16 (s. 6H); **^13^C NMR** (101 MHz, DMSO-*d_6_*) δ 168.2 (s), 163.7 (s), 160.8 (s), 159.6 (s), 156.6 (s), 145.8 (s), 143.3 (s), 139.0 (s), 137.5 (s), 130.7 (s), 128.7 (s), 127.6 (s), 126.4 (s), 125.7 (s), 125.0 (s), 124.5 (s), 123.2 (s), 118.3 (s), 114.4 (s), 107.0 (s), 99.1 (s), 63.8 (s), 47.8 (s), 45.2 (s), 27.1 (s)	**HRMS (ESI+)** calculated for [M + H]+ C_28_H_29_N_8_O_2_: 509.2408, found 509.2403.

## Data Availability

Data is contained within the article and Appendix A.

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
