# Peer review of "Design and Synthesis of Aminopyrimidinyl Pyrazole Analogs as PLK1 Inhibitors Using Hybrid 3D-QSAR and Molecular Docking"

_pharmaceuticals, 2022, doi:10.3390/ph15101170_

Round 1

Reviewer 1 Report (Previous Reviewer 1)

I revised this manuscript three times. Firstly, I recommended major revisions, then I suggested to authors to add some modifications to their revised manuscript. As authors followed the major part of my suggestions, I accepted the last version of revised manuscript.

 Now I agree with the authors: in my opinion is not necessary to synthesize all the compounds proposed in the manuscript. This is a computational study and additional synthesis and biological evaluation should be inserted in a future separate article.

For all these reasons, in my opinion this revised version of the manuscript can be accepted for publication

Author Response

I revised this manuscript three times. Firstly, I recommended major revisions, then I suggested to authors to add some modifications to their revised manuscript. As authors followed the major part of my suggestions, I accepted the last version of revised manuscript.

 Now I agree with the authors: in my opinion is not necessary to synthesize all the compounds proposed in the manuscript. This is a computational study and additional synthesis and biological evaluation should be inserted in a future separate article.

For all these reasons, in my opinion this revised version of the manuscript can be accepted for publication

; We thank the reviewer for accepting the manuscript after revision.

Reviewer 2 Report (Previous Reviewer 2)

Manuscript has been improved in revised resubmission and can be accepted as new synthesis of compound D39 and D40 has been included and spectral data is also provided but its better to include the NMR spectra of both compounds along with HRMS (ESI+) in supporting material of article.

Author Response

Reviewer 2
Manuscript has been improved in revised resubmission and can be accepted as new synthesis of compound D39 and D40 has been included and spectral data is also provided but its better to include the NMR spectra of both compounds along with HRMS (ESI+) in supporting material of article.-->1H NMR and 13C NMR spectra, and HRMS for both compounds, D39 and D40 are now added in the revised manuscript article and the copy of NMR spectra has been added in supplementary as suggested.

Reviewer 3 Report (Previous Reviewer 4)

The provided NMR data for D39 and D40 are questionable. Authors have to provide 1H and 13C NMR data for both compounds.

Author Response

The provided NMR data for D39 and D40 are questionable. Authors have to provide 1H and 13C NMR data for both compounds. --> 1H NMR and 13C NMR spectra, and HRMS for both compounds, D39 and D40 are now added in the revised manuscript article and the copy of NMR spectra has been added in supplementary as suggested.

Round 2

Reviewer 3 Report (Previous Reviewer 4)

Looking at the predicted IC50 values for D39 and D40 and calculated IC50 values, it can be clearly noticed the tremendous difference between both values. These results nullify the proposed virtual model for the PLK-1 inhibition study.

This manuscript is a resubmission of an earlier submission. The following is a list of the peer review reports and author responses from that submission.

Round 1

Reviewer 1 Report

The major part of my previous suggestions has been followed by the authors. in this revised form paper can be accepted for publication.

Reviewer 2 Report

Manuscript can be accepted in current form although its does not include experimental verification of results.

Reviewer 3 Report

An interesting knowledge has been reported, and also comments has been carried out carefully but however the following minor revision should be addressed before acceptance

Novelty of the manuscript must be better emphasized

The manuscript are not so surprising and informative, so The importance of the work should be highlighted in introduction section

1.      Typographical errors are present throughout the manuscript. Authors are required to pay keen attention to this

Reviewer 4 Report

The authors still rely only on computational methods without experimental validation.